# Self-actualization and B-values: Development and validation of two instruments in the Brazilian context

**Gustavo Henrique Silva de Souza**[1][☉]*, **Jorge Artur Peçanha de Miranda Coelho**[2][☉], **Nilton Cesar Lima**[3][‡], **Germano Gabriel Lima Esteves**[4][‡], **Fernanda Cristina Barbosa Pereira Queiroz**[5][‡], **Yuri Bento Marques**[1][‡]

1 Federal Institute of Northern Minas Gerais, Teófilo Otoni, Minas Gerais, Brazil, 2 Faculty of Medicine, Federal University of Alagoas, Maceió, Alagoas, Brazil, 3 Faculty of Accounting, Federal University of Uberlândia, Uberlândia, Minas Gerais, Brazil, 4 Faculty of Psychology, University of Rio Verde, Rio Verde, Goiás, Brazil, 5 Department of Production Engineering, Federal University of Rio Grande do Norte, Natal, Rio Grande do Norte, Brazil

☉ These authors contributed equally to this work.
‡ NCL, GGLE, FCBPQ and YBM also contributed equally to this work.
* gustavo.souza@ifnmg.edu.br

**Data Availability Statement:** The use and collection of data from human beings for academic research purposes, in accordance with Brazilian National Health Council resolutions 466/2012 and

## Abstract

Self-actualization is a complex psychological construct within Maslow's motivation theory, characterized by numerous gaps in the empirical and measurement spectrums. Therefore, the objectives of this study are to develop, validate, and cross-verify measures for self-actualization attributes and B-values, focusing on job context and theoretical congruence with innovative behavior and human values related to the self-actualization construct (suprapersonal subfunction). This study involved a diverse sample of 621 Brazilian participants from 25 different professions, indicating the broad applicability of the findings. The proposed instruments underwent content and semantic validity assessments, followed by verification of factor validity and internal consistency. Results showed satisfactory content, semantic and factor validity and internal consistency parameters. The study reveals that self-actualization attributes can be understood through achieving one's own potential and work meta-motivation, consistent with the adoption of B-values. Relationships with suprapersonal values (maturity, knowledge, and beauty) and innovative work behavior were also demonstrated, suggesting convergent validity evidence. The validation of SAAS and BVI contributes to understanding self-actualization and B-values in varied Brazilian contexts, offering insights for psychological assessment and intervention.

## Introduction

The hierarchy of needs (conative needs: physiological, safety, social, esteem, and self-actualization) in Maslow's motivation theory has presented limited evidence regarding the validity of its hierarchical structure [1]. Nevertheless, there is a consensus regarding its plausibility and internal theoretical consistency [2, 3]. Among the more intricate concepts comprising the

510/2016, requires express approval from an Ethics Committee. Furthermore, according to Circular Letter No. 2/2021/CONEP/SECNS/MS, in item 3.2., "once data collection is completed, the responsible researcher is recommended to download the collected data to a local electronic device, erasing any and all records from any virtual platform, shared environment or in the clouds". In other words, the data cannot be made available online and the transfer of this data must only occur upon approval. Therefore, all relevant data are within the manuscript and, if necessary, we can make the database available. The request must be sent to for the ethics committee responsible for approving the project, Human Research Ethics Committee of Faculdades Unificados de Teófilo Otoni (FUTO), by e-mail: etica.to@doctum.edu.br.

**Funding:** I declare that the author GHSdeS has received public funding from Instituto Federal do Norte de Minas Gerais (IFNMG, Brazil) (https://www.ifnmg.edu.br/), according to process no. 23791.000250/2020-38, in the amount of R$ 10,000.00. The funder had no role in study design, data collection and analysis, decision to publish, or preparation of the manuscript.

**Competing interests:** The authors have declared that no competing interests exist.

hierarchy of needs, self-actualization poses challenges in empirically operationalizing its understanding. This difficulty arises because, as per Maslow [4, 5], self-actualization covers highly specific experiences such as transcendence, meta-motivation, peak experiences, and prioritization of B-values (being values, a term referring to human values intrinsic to one's being). These experiences result from the fulfillment of all other lower-level basic needs and psychological well-being (or the absence of psychopathology, excluding psychosomatic illnesses such as stress or mild/moderate anxiety).

According to Maslow [4, 5], the central criteria for identifying self-actualization would be described as follows. (i) the absence of psychopathology (excluding psychosomatic illnesses such as stress or mild/moderate anxiety)–since mentally ill individuals often struggle to engage in meaningful social interactions, focus on personal and professional growth, and may not exhibit positive psychological functioning; in contrast, healthy individuals are more likely to self-develop. (ii) the fulfillment of basic needs following the progression of the hierarchy of needs—since individuals who satisfy basic needs are cognitively closer to self-fulfillment, fulfilling the fundamental requirements lays the foundations for individuals to pursue higher-level aspirations, ultimately leading to self-actualization; (iii) the adoption of B-values—since B-values align with essential characteristics and behaviors associated with individuals who have reached the pinnacle of personal development, the adoption of these values reflects a deeper understanding and appreciation of one's inner self, a commitment to personal growth and a sense of purpose that goes beyond materialistic pursuits, which are indicative of progress towards self-realization; and (iv) the utilization of one's own talents and individual capacities, along with the exploration of one's full potential in the job context—since self-actualized individuals realize and apply their abilities, promoting achievement and purpose, whose pursuit of career growth reflects a commitment to personal excellence and continuous self-improvement, fundamental aspects of self-actualization.

In practical terms, however, as per Maslow [6], self-actualization refers to a level of existential motivation in which individuals are inclined to fulfill their potential. Thus, the concept is directly linked to work—not in the sense of employment but in the sense of vocation (more approximate to a job), or what a human being is born to be and do [7]. More specifically, self-actualization could be defined as the need to become all that one is capable of, achievable only through the individual's actions. For instance, a musician must compose, an artist must paint, a poet must write, and a teacher must teach if they intend to be self-fulfilled and ultimately self-actualized [4]. In this way, self-actualization represents the fulfillment, excellence, and realization of one's unique individual idiosyncratic potential, much like the ability to become complete and happy with what one is capable of being [8].

Recent studies [9–11] report that individuals exhibiting self-actualization traits demonstrate meta-motivation linked to personal growth and overall well-being, including life satisfaction, love for humanity, self-acceptance, positive relationships, environmental concern, autonomy, life purpose, workplace engagement, creativity and innovative behavior, and self-awareness—in addition to experiences of transcendence. These characteristics show consistency with elements previously predicted by Maslow [4, 6, 8] and highlight common attributes for possible auto-actualization profiling.

In turn, the identification of self-actualized individuals or potential self-actualizers through measurement and psychometric testing has been the focus of several studies [10, 12–17] since the emergence of Maslow's motivation theory [18]. These studies tend to compile the key behaviors and characteristics directly or indirectly related to self-actualization. Although they associate self-actualization (in life as a whole) with job satisfaction—generally associated with innovative behavior [19]–, the attribution of the job as a motivator for (or resulting in) self-actualization is not firmly established in these studies [20].

Hence, unlike all other measures addressing self-actualization through generic personal characteristics focused on the individual's life/existence and B-values, this study assumes an intrinsic relationship between job and self-actualization. In this context, the nature of work influences the satisfaction and fulfillment of human beings with themselves. Therefore, the objectives of this study are to develop, validate, and cross-verify measures for self-actualization attributes and B-values, focusing on job context and theoretical congruence with innovative behavior and human values related to the self-actualization construct (suprapersonal subfunction).

This study presents the construction of the Self-Actualization Attributes Scale (SAAS), which encompasses emotions, attitudes, beliefs, and behaviors in the context of the job, with a particular focus on the influence of peak experiences and the state of flow; and the construction of the B-Values Inventory (BVI), which presents a list of 14 B-values in the form of observable behaviors—theoretically deemed priorities among self-actualizing individuals [8]. Previously, instruments such as the Personal Orientation Inventory (POI) [15] and the Characteristics of Self-Actualization Scale (CSAS) [10] have provided foundational insights but exhibit limitations in cultural applicability and comprehensive coverage of B-values, highlighting the need for the development of the SAAS and BVI tailored to the Brazilian context.

## Background and assumptions

Once the D-needs (deficiency needs—physiological, safety, social, and esteem needs) are satisfied, Maslow [4] posited that humans would then manifest B-needs (being needs or growth needs), which are cognitively more complex, sophisticated, and mature. B-needs are deemed secondary, typically activated after the fulfillment of D-needs, and they may strengthen progressively with the complete satisfaction of more basic needs.

According to Maslow [5], B-needs encompass cognitive needs (e.g., knowledge, meaning, and self-awareness), aesthetic needs (e.g., beauty, balance, and form), and self-actualization needs. The term "self-actualization" was originally coined by Goldstein [21], describing processes of self-development of individual capacities, talents, and potentialities. The concept reflects the Aristotelian notion proposed in the Act and Potency theory [22], where potency is what an individual can become, do, or produce, and act is the development of that potency to the utmost, fulfilling the role and purpose of one's existence, leading the individual toward eudaimonia (a state of continuous contentment, towards fulfillment) [23].

Therefore, the need for self-actualization entails personal and professional growth and self-development related to the expertise and talents of each individual, whose pursuit is to fulfill oneself. It involves self-awareness, signifying that self-actualization can only be achieved through something that an individual can do for themselves. Generally, it translates into what the person does, produces, creates, develops, executes, etc. [4].

On the other hand, Maslow [8] explored an integrated concept between self-actualization and individual high-level motivation, meta-needs, whose argument revolved around psychological health—akin to individuation. According to the author, metaneeds could be psychologically configured as specific values, called B-values.

B-values can be identified as expressions of a human being who is in their fullness and maturity, prioritizing a meaningful life free from psychopathologies—the self-actualizers [4]. Studies by Klisanin [24] and Kossewska and Potměšil [25], for example, have associated the adoption of B-values as a personal orientation to alleviate social, psychological, and educational issues. Consequently, advancing in the hierarchy of needs would be a sine qua non condition for reaching this level of motivation. B-values would be interrelated, representing 14 prioritized values that merge and overlap: truth, playfulness, aliveness, perfection, goodness,

justice, beauty, simplicity, richness, uniqueness, effortlessness, completion, self-sufficiency, and wholeness [4].

In addition to B-values, Maslow [4] identified that self-actualizing individuals exhibited a strong sense of creativity (expressive and spontaneous) that led them to peak experiences. Peak experiences translate into moments when an individual engages in a unique activity or action (whether at the level of abstraction, sensory observation, or concrete experience) that generates a kind of catharsis (or climax), akin to a sense of exhilaration. During or after the experience, the individual feels happy, fortunate, and fulfilled—as if at the height of their powers—in an intrinsically self-justified moment that self-validates itself. Commonly, during these peak experiences, the related activity is prioritized to the extreme, and individuals exhibit disorientation in time and space, losing track of hours and other needs and commitments.

Moreover, peak experiences are directly associated with individual identity, generally representing the closest one can define an individual (who they truly are) [4]. Regarding this, engaging in activities where one's talent is indispensable tends to generate a greater quantity and quality of peak experiences [8], aligning strongly with the concept of fulfilling work, as a profession (job), or what one was born to be.

Peak experiences during engagement in fulfilling work (preferred activities or those linked to individual potentialities) have been associated with the term "optimal experience" based on the concept of flow. The flow state implies an individual's involvement in a specific task with a defined goal, characterized by high immersion, control, and concentration. This generates a sense of intrinsic satisfaction and well-being caused by the task itself (autotelic activity), making it seem at that moment that nothing else matters, akin to abstraction [26]. The flow state is commonly identified in artists, athletes, gamers, musicians, chess players, surgeons, writers, and professors [27–34]. Furthermore, the moment of experiencing flow constitutes a high-performance behavior characterized by consistency and excellence in task and activity execution. It brings about a sense of loss of self-awareness and a transformation of the perception of time [28, 35]. Time appears to pass more swiftly and with enhanced quality.

Thus, despite empirically seeming to resemble optimal experiences, peak experiences are characteristic sensations of a cognitive pinnacle—even of transcendence—whose process may involve the occurrence of preeminent flow experiences concomitantly. Therefore, self-actualization as an axiomatic construct for the motives that make it a necessity may imply well-being (joy, ecstasy, or transcendence) generated by a specific task. It makes the performed task the element that fulfills its need, much like food satisfies a physiological need or praise fulfills a need for esteem.

## Methods

### Part 1—Operationalization, construction, and evaluation of items for the proposed instruments

**Self-Actualization Attributes Scale (SAAS).** The initial step in constructing the SAAS involved defining a structure for empirical operationalization (source of items) based on self-actualization attributes linked to the job, taking into consideration the influence of peak experiences [4] and the state of flow [28]. Within this framework, 43 items were developed to identify emotions, attitudes, beliefs, traits, and behaviors characteristic of self-actualized individuals or those in the process of self-actualization (e.g., "I seek to give my best to my job, exploring all my potential", "when I am working, I lose track of time", or "I feel satisfied with what I have become professionally"). The formulation of items adhered to the criteria recommended by Pasquali [36] and Cohen and Swerdlik [37].

The focus on job distinguishes the SAAS from other measures related to self-actualization [10, 12–17]. Due to this theoretical and methodological choice, some items in the SAAS exhibit content alignment with other concepts in the work context that reflect on health and personal development in the workplace, such as work engagement [38], well-being, and job satisfaction [39, 40], and the meaning of work [41].

**B-Values Inventory (BVI).** For the construction of the BVI, items were formulated for each of the 14 B-values determined by Maslow [4], considering the closest description of the cognitive and behavioral representation that these values encompass. Correspondingly, Shostrom's Personal Orientation Inventory (POI) [15, 16] measures values (including B-values) and attitudes using a dichotomous scale, encompassing the hierarchy of needs and aiming to identify self-actualized individuals. However, due to issues in its psychometric structure, Weiss [42] does not recommend its use for clinical or research purposes. On the other hand, Kaufman's Characteristics of Self-Actualization Scale (CSAS) [10] includes various B-values but does not cover all 14 core values. Therefore, the choice was made to represent the 14 B-values as observable self-description variables, such as "Truth. True person who avoids telling lies", or "Aliveness. Person full of energy, with vigor and spontaneity".

**Content validity of the instruments.** Content validity of the instruments was conducted through the analysis of 7 (seven) judges, including 2 psychometricians (1 Ph.D. in social psychology and 1 Ph.D. in social, work, and organizational psychology), 4 professors in the field of organizational behavior (1 Ph.D. in production engineering, 2 Ph.Ds. in business administration, and 1 Ph.D. candidate in business administration), and 1 specialist in organizational behavior (psychologist and human resources management specialist). Initially, the judges were presented with the respective constructs' definitions for each instrument: self-actualization with a focus on work (for the SAAS), which includes the individual achievement of one's potential and work meta-motivation; and a description of the B-values defined by Maslow [4] (for the BVI). The judges were asked to indicate the degree of appropriateness of the content of each item concerning the representation of the associated behavior or characteristic, considering the criteria established by Hernández-Nieto [43] and Pasquali [36] regarding theoretical relevance, practical relevance, and language clarity.

In the analysis of the SAAS, out of the 43 evaluated items, three showed inadequacy and were removed, leaving 40 items for further analysis. Additionally, specific changes were suggested; for example, for the item "I seek to be all that I am capable of being", a modification to "I seek to be all that I am capable of being in my work" was proposed and accepted by the researchers. For the BVI, all 14 items were endorsed as appropriate, with no additional suggestions.

**Semantic validity of the instruments.** Semantic validity of the instruments was conducted through an intelligibility analysis of the items, aiming to verify if the constructed items express the item's content intelligibly and without leaving doubts of interpretation for the target audience [36]. For this stage, input was gathered from 3 professionals with high school education, 2 university students, and 3 professors in the field of languages and literature. All items from the SAAS and BVI were endorsed as appropriate for composing the initial version of the proposed instruments.

## Part 2—Empirical study: Psychometric aspects of the proposed instruments

**Procedures / participants recruitment.** The instruments were administered through accessibility, utilizing a non-probabilistic convenience sampling method [44]. Participants were individually invited via email to respond voluntarily to the instruments through an online

questionnaire between June 29 and August 31, 2020. We sent emails to professionals from the contact networks of the authors of this study and other research partners and colleagues. The response rate from participants was 11.96% of all 5,190 emails sent. The inclusion criterion for the study population was people who had a job. Retired and unemployed persons must be excluded; however, there were no exclusions. Also, there was no missing data, as the online questionnaire application system did not allow participants to send answers that contained missing data.

**Participants.**   The empirical study involved 621 Brazilian participants from 25 different professions, with the majority being professors (63%) and administrators (5%)—it is in line with the interest of this study, since the teaching profession is commonly related to individual talents and job satisfaction [27, 31], and probably one of the types of target population for the instruments that have been developed—, of whom 55.7% were female, with an average age of 42 years (range from 18 to 81 years; SD = 15.02). Participants came from 24 Federative States of Brazil, with the most representative states being Minas Gerais (30.9%), São Paulo (18.1%), and Paraná (12.5%).

**Instruments.**   621 Brazilian participants were required to respond to five research instruments in Portuguese language (S1 File), namely: (1) Self-Actualization Attributes Scale (SAAS), (2) B-Values Inventory (BVI), (3) Innovation Potential Scale (IPS), (4) Basic Values Questionnaire (BVQ), and (5) Sociodemographic Questionnaire.

1. Self-Actualization Attributes Scale (SAAS): It specifically developed for this study to identify characteristics and behaviors related to self-actualization. Participants responded to the 40-item SAAS in a shuffled Likert scale ranging from 1 = Does not Describe me at All to 5 = Describes me Completely.

2. B-Values Inventory (BVI): Constructed for this study, listing the 14 B-Values determined by Maslow [4] with additional descriptions as characteristics or attitudes in response to the phrase "Generally, I feel that I am a person...". The BVI used a 5-point Likert scale ranging from 1 = Does not Describe me at All to 5 = Describes me Completely.

3. Innovation Potential Scale (IPS): A self-report psychometric test with a 5-point scale (ranging from 1 = Does not Describe me at All to 5 = Describes me Completely), consisting of 9 items. The instrument assesses characteristics of an innovative, creative, and proactive person within the work context, including items such as "I innovate in my work to become more productive" and "I am a creative person". IPS demonstrates factor validity and internal consistency in a one-factor model titled "Innovation Potential in the Work Context" [45].

4. Basic Values Questionnaire (BVQ): A psychometric self-report test with a 7-point scale (ranging from 1 = Not Important at All to 7 = Extremely Important), comprising 18 basic values and their descriptions. The BVQ presents confirmatory factor validity in a model of 6 value subfunctions (Interactive, Suprapersonal, Excitement, Experimentation, Normative, Existence, and Promotion), with acceptable fit indices [$\chi2$ (df) = 190.36 (5); GFI = 0.900; CFI = 0.91; RMSEA = 0.08], including items such as: "Maturity (Feeling that you have achieved your goals in life; developing all your capacities)" and "Success (Achieving what you set out to do; being efficient in everything you do)" [46].

5. Sociodemographic Questionnaire: It used to understand and characterize the sample, including items such as gender, age, place of residence, and profession.

**Ethics approval.** We strictly adhere to the Brazilian National Health Council resolutions 466/12 and 510/16. Furthermore, we strictly adhered to all of the standards outlined in Circular Letter No. 2/2021/CONEP/SECNS/MS. This research investigation was approved by the Human Research Ethics Committee of Faculdades Unificados de Teófilo Otoni (FUTO) (Protocol No. 31508720.8.0000.8747). As a result, the participants were informed that they were not required to participate in the research and that they might withdraw at any time if they were uncomfortable for any reason (voluntary participation was ensured). Before answering the questionnaire, the participants (every participant was over the age of 18) agreed to an Informed Consent Form (ICF) electronically signed, in which we informed the participants that their provided information would be confidential, with no way of individual identification, and that their responses would be analyzed as a whole rather than individually. Each participant spent around twelve minutes reading the ICF and answering the questionnaire.

**Data analysis.** The collected data were tabulated and processed using PSPP (free software from GNU), which was employed for descriptive statistical analyses. The Factor version 12.01.02 (freeware program) was used for Exploratory Factor Analysis (EFA), and for estimating quality of fit indices and additional indicators of dimensionality, predictive effectiveness, reliability, and factor determination [47]. Following criteria indicated by Ferrando & Lorenzo-Seva [47], Factor Determinacy Index (FDI) above 0.90, ORION marginal reliabilities above 0.80, sensitivity rates above 2, and expected percentages of true differences above 90% are recommended.

EFA was applied to the instruments constructed in this study using the polychoric correlation matrix, with the Robust Diagonally Weighted Least Squares (RDWLS) extraction method for SAAS and Robust Unweighted Least Squares (RULS) for BVI [48]. The RDWLS and RULS methods were selected for their robustness in handling ordinal data and their suitability for the distribution characteristics of the survey responses, ensuring accurate factor extraction and reliability assessments in the context of psychometrics scale validation. Parallel Analysis determined the decision about the number of factors to be retained [49]. To verify evidence of internal consistency of the instruments constructed in this study, Cronbach's alpha coefficient ($\alpha$), the Composite Reliability (CR) coefficient, and McDonald's Omega ($\omega$) were calculated. Factor stability was assessed through the H-index. The H-index evaluates how well a set of items represents a common factor. H values range from 0 to 1. High values of H ($> 0.80$) suggest a well-defined latent variable, meaning it is likely to be stable across different studies [47].

Model fit was evaluated using Root Mean Square Error of Approximation (RMSEA), Comparative Fit Index (CFI), and Tucker-Lewis Index (TLI). RMSEA values should be less than 0.08, with values equal to or less than 0.05 indicating excellent fit. CFI and TLI values should be above 0.90, or preferably 0.95, to indicate a valid model. For Chi-square/degrees of freedom ($\chi2$/df), values less than 2 indicate an excellent fit, values between 2 and 3 suggest a good fit, and values approximate to 5 indicate an acceptable fit [50, 51]. Additionally, one-factor solutions were analyzed from Unidimensional Congruence (UniCo), Explained Common Variance (ECV) and Mean of Item REsidual Absolute Loadings (MIREAL). UniCo larger than 0.95, ECV larger than 0.85 and MIREAL lower than 0.30 suggest that data can be treated as essentially unidimensional [47].

In the search for evidence of theoretical-empirical congruence among the investigated constructs, according to the objectives of this study, the mean factor score of the applied instruments was used to conduct Spearman's Rho correlation tests. Thus, the convergence between factors measuring self-actualization attributes, B-values, innovation in the work context, and the suprapersonal values (which involves values theoretically and empirically related to self-actualization: knowledge, beauty, and maturity) was verified.

## Results

### Parameters of validity and internal consistency of the SAAS

Initially, the factorability of the SAAS matrix was confirmed through the Kaiser-Meyer-Olkin test (KMO = 0.94) and Bartlett's test of sphericity ($\chi2$(703) = 7235.90; p = 0.00). To determine the number of factors in the correlation matrix, the Parallel Analysis was employed [49], indicating a two-factor model (Real-data eigenvalues: 19.12 and 2.03) with a total explained variance of 59.70%. Given these results, EFA was conducted using the polychoric correlation matrix, employing the RDWLS extraction method [48]. The analysis fixed the number of factors at two and utilized Promax rotation. From the analysis, one item (Item 01) was excluded due to a factor loading below 0.30 on all factors, and four items (Items 05, 31, 33, and 38) were excluded due to cross-loaded on both factors, as recommended in the literature [52]. The exclusion of these items suggests a need to refine the conceptual alignment of the SAAS's dimensions. Future iterations could explore alternative phrasings or conceptual frameworks to ensure all aspects of self-actualization are accurately captured.

Following this step, considering the exclusion of 5 items, a decision was made to conduct the Parallel Analysis [49] once again, confirming the two-factor model. EFA was then performed again on the revised SAAS (with 35 items). The analysis fixed the number of factors at two and utilized Promax rotation. Factor loadings for the items on their respective factors are presented in Table 1, along with the values of Cronbach's alpha, Composite Reliability coefficient, and H-Latent.

The two factors extracted from the SAAS exhibited satisfactory internal consistency and determination index. Based on item analysis, Factor 1 (composed of items 4, 22, 23, 24, 26, 27, 30, and 32) can be labeled as "Achievement of the Own Potential", representing the pursuit of the individual's apex in terms of evolution, knowledge, personal and professional excellence, self-awareness, aiming to achieve their fullest potential. On the other hand, Factor 2 (composed of items 2, 3, 6 to 21, 25, 28, 29, 34 to 37, 39, and 40), by item analysis, can be labeled as "Work Meta-motivation", representing work driven by a sense of purpose, well-being, and satisfaction intrinsic to the essence of work. It is motivated by the representation of the individual's nature, grounded in growth values. Furthermore, the H-Latent index—indicating how replicable the factors are in future studies (H ≥ 0.80)–showed that Factor 1 (H = 0.98) and Factor 2 (H = 0.94) are replicable, as indicated by Hancock & Mueller [53]. To demonstrate the quality and effectiveness of the proposed factor structure for the SAAS, additional sensitivity and reliability indicators, along with fit quality indices for the two-factor model, are presented in Table 2.

The two-factor model exhibited acceptable fit indexes—CFI and TLI above 0.95, and RMSEA less than 0.05 –, along with sensitivity, reliability, and factor determination indicators that support the psychometric quality of the instrument and emphasize its factor structure (valid model), as recommended in the literature [49, 51]. The study suggests a comprehension of the SAAS as a two-factor measure that assesses self-actualization from the perspective of fulfilling work. In practical terms, the psychometric quality indicators indicate the potential for the instrument's use and replication, considering the measurement of achieving one's own potential and work meta-motivation.

### Parameters of validity and internal consistency of the BVI

For BVI, the factorability of the correlation matrix was accepted in accordance with the Kaiser-Meyer-Olkin test (KMO = 0.80) and the Bartlett's sphericity test ($\chi2$(91) = 3052.80; p = 0.00). The identification of the number of factors in the correlation matrix was performed

**Table 1. Factor loadings of items from SAAS and reliability indexes.**

| | Items | Factor 1 | Factor 2 |
|---|---|---|---|
| 27 | I seek knowledge to improve myself every day. | **1.04** | -0.17 |
| 30 | I aim to evolve and develop myself. | **0.90** | -0.02 |
| 24 | I strive to dedicate myself fully to my job, exploring my full potential. | **0.78** | 0.04 |
| 4 | I pursue professional (technical) excellence in my job. | **0.77** | -0.05 |
| 23 | I seek personal excellence in my job. | **0.71** | 0.08 |
| 22 | I aim to be everything I am capable of being in my work. | **0.71** | 0.06 |
| 32 | I consistently strive for excellence in myself. | **0.62** | 0.06 |
| 26 | I am aware of my potential. | **0.45** | 0.12 |
| 19 | Executing my job brings me well-being. | -0.08 | **0.96** |
| 17 | Among other things, my job gives meaning to my life. | -0.11 | **0.94** |
| 10 | Achievements related to my job are a source of happiness. | -0.06 | **0.89** |
| 7 | My job is intrinsic to my own nature. | -0.13 | **0.87** |
| 18 | My job reflects who I truly am. | -0.07 | **0.87** |
| 35 | I am passionate about what I do. | 0.03 | **0.86** |
| 13 | I feel proud of my job. | -0.03 | **0.86** |
| 12 | I eagerly look forward to carrying out my job. | -0.07 | **0.86** |
| 20 | My professional life has purpose. | 0.05 | **0.85** |
| 25 | I feel fulfilled by what I do. | 0.06 | **0.84** |
| 28 | Upon completing my job, I feel that I could continue this activity for many years due to the satisfaction it brings. | 0.02 | **0.81** |
| 8 | I forget about everyday problems when I am working. | -0.21 | **0.79** |
| 6 | I see beauty in the job I perform. | 0.02 | **0.77** |
| 11 | I enjoy discussing what I am working on with others. | -0.01 | **0.76** |
| 14 | At times, I feel euphoric when engaged in my work. | -0.03 | **0.74** |
| 29 | I feel satisfied with what I have become professionally. | 0.09 | **0.74** |
| 2 | When working, I am having fun. | -0.10 | **0.71** |
| 39 | My profession is aligned with my talents. | 0.05 | **0.70** |
| 36 | Upon completing my work, I feel that I have played my role in the world. | 0.13 | **0.68** |
| 34 | I feel spiritually connected to my work. | 0.06 | **0.67** |
| 37 | My professional performance enhances my self-esteem. | 0.09 | **0.67** |
| 9 | I know which job/profession personally fulfills me. | 0.02 | **0.67** |
| 21 | My job is a very important part of my life. | 0.16 | **0.63** |
| 16 | I don't mind working extra hours in my job. | 0.00 | **0.58** |
| 40 | When I am working, I lose track of time. | 0.03 | **0.57** |
| 15 | I know that others recognize me for what I do in my job. | 0.05 | **0.48** |
| 3 | I believe my job can change the world or its paradigms. | 0.23 | **0.36** |
| | **Number of Items** | 8 | 27 |
| | **Real-data eigenvalues** | 18.59 | 2.30 |
| | **Explained Variance** | 53.13% | 6.57% |
| | **Cronbach's Alpha** | 0.86 | 0.96 |
| | **Composite Reliability** | 0.91 | 0.97 |
| | **McDonald's Omega** | 0.95 | 0.95 |
| | **H-*Latent*** | 0.98 | 0.94 |

Source: research data.

**Table 2. Model fit indexes and indicators of sensitivity, reliability, and factor determination.**

| Indexes | two-factor model | |
|---|---|---|
| CFI | 0.99 | |
| TLI | 0.99 | |
| RMSEA (90% CI) | 0.04 (0.03–0.04) | |
| Qui-Square ($\chi^2$) | 1251.60 | |
| Degree Free (df) | 526 | |
| *p-value* | $< 0.001$ | |
| Ratio $\chi^2$/gl | 2.37 | |
| *Factor Determinacy Index* (FDI) | 0.99[i] | 0.97[ii] |
| ORION (marginal reliability) | 0.98[i] | 0.94[ii] |
| *Sensitivity ratio (SR)* | 7.02[i] | 4.20[ii] |
| *Expected percentage of true differences* (EPTD) | 97.90%[i] | 95.10%[ii] |

Source: research data.

Notes:

[i] Factor 1—Achieving of own potential;

[ii] Factor 2—Work Meta-motivation.

through Parallel Analysis [49], indicating a one-factor model (Real-data eigenvalue: 4.07) and explained variance of 38.51%. EFA was conducted using the polychoric correlation matrix, with the Robust Unweighted Least Squares (RULS) extraction method [48]. All 14 items loaded onto a single factor (see Table 3).

The factor solution indicated that the 14 B-values present a one-factor structure, with factor loadings ranging from 0.30 to 0.68, with satisfactory indicators of internal consistency. Additionally, the H-Latent pointed to the replicability [53] of the overall factor of B-values (H = 0.87). In addition, complementary analyses corroborated a factor structure for BVI and fit indexes of the one-factor model (see Table 4). The one-factor model showed acceptable fit indexes—CFI and TLI above 0.95, and RMSEA less than 0.05, according parameters from the literature [50, 51]–, along with sensitivity, reliability, and factor determination indicators that support the psychometric quality of the instrument and emphasize its factor structure (valid model). Unidimensionality coefficients and the factor determiner index corroborate the one-dimensional structure.

The study indicates an understanding of BVI as a unidimensional measure that consolidates the 14 B-values determined by Maslow [4] into a single dimension. In practical terms, psychometric quality indicators suggest the potential use and replication of the instrument, considering the measurement of all 14 B-values together to represent the prioritized values of self-actualized individuals.

## Additional evidence of theoretical-empirical congruence

In order to verify additional evidence of theoretical-empirical congruence among the investigated constructs, a convergence analysis (through Spearman's correlation) was proposed between self-actualization attributes, B-values, innovative behavior in the work context, and human values. Table 5 presents the mean and standard deviation, as well as correlation analyses between the factors of SAAS, BVI, IPS, and the suprapersonal subfunction (BVQ).

Overall, there are moderate to strong positive correlation—interpretation based on the literature [44]–, with p values below 0.05, among the investigated constructs, emphasizing the tendency for convergence between them. Initially, it is worth noting that the correlation

**Table 3. Factor loadings of items from BVI and reliability indexes.**

|  | Items | Factor Loadings |
|---|---|---|
| 3 | Aliveness. Person full of energy, with vigor and spontaneity. | 0.68 |
| 9 | Richness. Complete and fulfilled person who seeks the height of its fullness. | 0.66 |
| 14 | Wholeness. A person of integrity who prioritizes seriousness, honesty and dignity. | 0.65 |
| 6 | Justice. Person with a sense of justice who prioritizes order and merit. | 0.63 |
| 2 | Playfulness. Cheerful person who prioritizes good humor. | 0.61 |
| 5 | Goodness. Kind person who prioritizes altruism and benevolence. | 0.61 |
| 12 | Completion. Accomplished person who undertakes significant tasks and fulfilling one's destiny. | 0.58 |
| 8 | Simplicity. Person inclined to simplicity, who cares about the simple things in life. | 0.53 |
| 10 | Singularity. Original person who seeks to be unique and exclusive in its individuality. | 0.51 |
| 1 | Truth. True person who avoids telling lies. | 0.50 |
| 4 | Perfection. Meticulous person who strives for excellence and perfection. | 0.49 |
| 13 | Self-sufficiency. Self-sufficient person who seeks to have autonomy and independence in what does. | 0.41 |
| 7 | Beauty. Person inclined towards art who prioritizes beauty and aesthetics. | 0.31 |
| 11 | Effortlessness. Thrifty person who aims to accomplish important tasks with efficiency and low effort. | 0,30 |
| **Number of Items** | | 14 |
| **Real-data eigenvalues** | | 4.07 |
| **Explained Variance** | | 38.51% |
| **Cronbach's Alpha** | | 0.84 |
| **Composite Reliability** | | 0.85 |
| **McDonald's Omega** | | 0.84 |
| **H-*Latent*** | | 0.86 |

Source: research data.

**Table 4. Model fit indexes and indicators of sensitivity, reliability, and factor determination.**

| Indexes | one-factor model | |
|---|---|---|
| CFI | 0.93 | |
| TLI | 0.92 | |
| RMSEA (90% CI) | 0.08 (0.07–0.08) | |
| Qui-Square ($\chi^2$) | 437.27 | |
| Degree Free (df) | 77 | |
| *p-value* | < 0.001 | |
| Ratio $\chi^2$/gl | 5.67 | |
| Unidimensionality Coefficients | uniCo | 0.96 |
|  | ECV | 0.79 |
|  | MIREAL | 0.23 |
| Factor Determinacy Index (FDI) | 0.92 | |
| ORION (marginal reliability) | 0.86 | |
| Sensitivity ratio (SR) | 2.49 | |
| Expected percentage of true differences (EPTD) | 90.80%[ii] | |

Source: research data.

**Table 5. Descriptive and correlation analysis between the factors from SAAS, BVI, IPS and BVQ.**

|  | B-values | Achievement of the Own Potential | Work Meta-motivation | Innovation in the Work Context | Suprapersonal Subfunction |
|---|---|---|---|---|---|
| Mean (SD) | 3.98 (0.46) | 4.37 (0.54) | 3.94 (0.74) | 3.86 (0.70) | 5.93 (0.87) |
| Achievement of the Own Potential | 0.60** |  | 0.75** | 0.42** | 0.37** |
| Work Meta-motivation | 0.53** | 0.75** |  | 0.33** | 0.24** |
| Innovation in the Work Context | 0.46** | 0.42** | 0.33** |  | 0.34** |
| Suprapersonal Subfunction | 0.40** | 0.37** | 0.24** | 0.34** |  |

Source: research data.

** $p > 0.001$

analyses show that the set of B-values moderately correlated with (i) Achievement of the Own Potential (Factor 1 of SAAS), (ii) Work Meta-motivation (Factor 2 of SAAS), (iii) innovative behavior in the work context (IPS), and (iv) the Suprapersonal subfunction (representing the central-humanistic dimension of BVQ). The factors of SAAS (Achievement of the Own Potential and Work Meta-motivation) also showed a strong correlation with each other (rho = 0.75). Innovative behavior in the work context and Achievement of the Own Potential also exhibited a moderate correlation with each other (rho = 0.43).

## Discussion

The study initially introduces the development and validation of two distinct instruments: the SAAS (attributes of self-actualization through work) and the BVI (B-values). However, theoretically, these instruments are structurally related since, according to Maslow [4], self-actualized individuals would adopt (prioritize) B-values (e.g., aliveness, completion, uniqueness, and self-sufficiency).

The SAAS presented a two-factor structure with evidence of adequate psychometric parameters regarding factor validity and internal consistency. The two-factor structure is grounded in "Achievement of the Own Potential" (Factor 1), reflecting individual efforts toward personal growth, and "Work Meta-motivation" (Factor 2), expressing the role of fulfilling work (profession, in the broader sense, or what the individual was born to be and do, in a cosmogonic sense) as an intrinsic motivational element.

On the other hand, the BVI presented a one-factor structure with evidence of adequate psychometric parameters regarding factor validity and internal consistency. The unidimensional structure aligns with Maslow's [4] theoretical model, which posited that the B-values would be interrelated as an expression of an individual in their fullness and maturity, thus representing self-actualized individuals.

The SAAS fills a gap in research on self-actualization by directly linking it to the job, particularly by encompassing peak and flow experiences during the work process. Additionally, the BVI systematizes the 14 B-values into scalable items, roughly adhering to Maslow's [4] theoretical proposition. In Brazilian cultural contexts, where job satisfaction and personal fulfillment are deeply interwoven with community and social ties, the SAAS and BVI can offer nuanced insights into how self-actualization and B-values manifest in professional settings. For instance, in education and healthcare sectors, where altruism and community service are prevalent, these instruments can help identify specific motivations and values driving job satisfaction and personal growth.

In addition to assessing the psychometric parameters of the developed instruments, we tested the correlation among the factors of SAAS, BVI, innovation in the work context (a unidimensional factor of IPS), and the suprapersonal subfunction (a factor in BVQ related to self-actualization). The congruence among self-actualization attributes, values-B, innovation in the work context, and suprapersonal values highlights a theoretically consistent alignment with motivational theory and the assumptions advocated by Maslow [4, 7, 8]. Firstly, it supports the proposal that self-actualization (specifically attributed to work in this case) is related to B-values [4]. Secondly, it endorses the idea that achieving one's own potential and B-values are related to maturity, knowledge, and beauty values—constituting the suprapersonal evaluative subfunction, representing humanitarian motivators and central orientation, and encompassing aesthetic, cognitive, and self-actualization needs [46]. Thirdly, it supports the notion that self-actualization is related to increased creative behavior (here assessed in elements of innovation in the work context)—see Maslow [7, 8].

In functionalist terms, tools focused on self-actualization through work enable a specific analysis of the individual work context, bordering on quality of life and job satisfaction, while understanding the nature of work as an intrinsic motivator. This provides data that can inform decisions and human resource management policies, especially concerning competency-based management. Additionally, it aids theoretical and empirical studies on the subject of self-actualization.

## Conclusions, limitations and future research

This study aimed to develop and verify evidence of the validity and internal consistency of two measures related to self-actualization, the Self-Actualization Attributes Scale (SAAS) and the B-Values Inventory (BVI), in addition to verify evidence of theoretical-empirical congruence among the self-actualization, the adoption of B-values, the innovative behavior in the work context, and the human values. In this study, we developed the aforementioned instruments and presented their psychometric parameters of content, factorial, construct and convergent validity for the Brazilian context.

Therefore, the SAAS presents applicability and effectiveness in identifying whether an individual achieves self-actualization through their job, although it may not necessarily indicate whether the individual is inherently self-actualized. Fulfilling and self-motivating activities may not always be directly associated with employment or job roles. This consideration highlights the importance of psychological assessment to understand the practical implications of self-actualization measures in different contexts. It highlights the relevance of evaluating self-actualization attributes in conjunction with B-values, with a view to a more concrete definition of the potential for self-actualization. Recommendations for practical and clinical application, including integrated self-actualization measures into screening and assessment protocols, designing tailored interventions, and fostering cross-disciplinary collaboration, remain crucial for maximizing their impact in fields like education, healthcare, and organizational development.

As limitations of this study, we initially highlight the theoretical-empirical approach focused on fulfilling work, which may not apply to all individuals and their respective occupations. In this study, despite the convenience sampling approach, the majority of research participants fell within the profession of teaching (63%), which is a field typically characterized by a direct connection to one's vocation and inherent talents and, therefore, essential for establishing the prerequisites of work-associated self-actualization—this pattern is also expected in professions related to art, literature, science, and sports [54–57]. Specifically, about professions and other sociodemographic variables, the lack of stratification of the sample is a limitation of the study. Although the study included individuals from various professions, beyond the

categories of professors (63%) and administrators (5%), there is not a sufficient minimum quantity to test differences between the identified professions.

Another limitation of the study lies in the methodological approach, which primarily focused on testing and verifying the psychometric quality parameters of the developed instruments. Consequently, the association of the SAAS and the BVI with other potentially underlying instruments or constructs was not tested beyond the presented correlation analyses. Thus, the study provides only evidence related to the construct.

Therefore, for future research agendas, it is suggested to conduct additional analyses between the SAAS and the BVI concerning human values [46], the meaning of work [41], well-being [40], and flow experiences [28]. Additionally, procedures for convergent validity are recommended, using the Brief Index of Self-Actualization [17], for example. Comparative studies among professions could provide predictive validity parameters. Longitudinal studies and cultural adaptation for the instruments developed here are also necessary to establish more effective measurements for self-actualization in different contexts.

Specifically, for the SAAS, improvement possibilities for the instrument include testing shortened versions that can measure the construct with fewer observable variables. For both instruments (SAAS and BVI), testing with new samples is necessary to obtain additional evidence of their validity and data to support their standardization. The validation of SAAS and BVI for the Brazilian context opens avenues for their application in organizational psychology, career counseling, and personal development programs. Future research should explore longitudinal studies to assess how self-actualization and B-values evolve over time and across different life stages and professional paths.

## Supporting information

**S1 File. Instruments.**
(ZIP)

## Acknowledgments

The authors would like to thank IFNMG for supporting the research that generated the data in this paper.

## Author Contributions

**Conceptualization:** Gustavo Henrique Silva de Souza.

**Data curation:** Gustavo Henrique Silva de Souza, Nilton Cesar Lima, Germano Gabriel Lima Esteves, Fernanda Cristina Barbosa Pereira Queiroz.

**Formal analysis:** Gustavo Henrique Silva de Souza, Jorge Artur Peçanha de Miranda Coelho, Nilton Cesar Lima.

**Funding acquisition:** Gustavo Henrique Silva de Souza.

**Investigation:** Gustavo Henrique Silva de Souza, Nilton Cesar Lima, Germano Gabriel Lima Esteves, Fernanda Cristina Barbosa Pereira Queiroz, Yuri Bento Marques.

**Methodology:** Gustavo Henrique Silva de Souza, Jorge Artur Peçanha de Miranda Coelho, Germano Gabriel Lima Esteves, Yuri Bento Marques.

**Project administration:** Gustavo Henrique Silva de Souza, Nilton Cesar Lima, Yuri Bento Marques.

**Resources:** Gustavo Henrique Silva de Souza, Fernanda Cristina Barbosa Pereira Queiroz.

**Software:** Gustavo Henrique Silva de Souza, Jorge Artur Peçanha de Miranda Coelho, Germano Gabriel Lima Esteves, Yuri Bento Marques.

**Supervision:** Gustavo Henrique Silva de Souza, Jorge Artur Peçanha de Miranda Coelho, Nilton Cesar Lima, Fernanda Cristina Barbosa Pereira Queiroz.

**Validation:** Gustavo Henrique Silva de Souza, Jorge Artur Peçanha de Miranda Coelho, Germano Gabriel Lima Esteves.

**Visualization:** Gustavo Henrique Silva de Souza, Jorge Artur Peçanha de Miranda Coelho.

**Writing – original draft:** Gustavo Henrique Silva de Souza, Jorge Artur Peçanha de Miranda Coelho.

**Writing – review & editing:** Gustavo Henrique Silva de Souza, Jorge Artur Peçanha de Miranda Coelho, Nilton Cesar Lima, Fernanda Cristina Barbosa Pereira Queiroz, Yuri Bento Marques.

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
