## [Decision Letter · Decision Letter 0]

19 Feb 2024

PONE-D-24-00059Measuring Self-Actualization and B-Values: Construction and Validation of Two Instruments in the Brazilian ContextPLOS ONE

Dear Dr. Souza,

Thank you for submitting your manuscript to PLOS ONE. After careful consideration, we feel that it has merit but does not fully meet PLOS ONE’s publication criteria as it currently stands. Therefore, we invite you to submit a revised version of the manuscript that addresses the points raised during the review process.

We look forward to receiving your revised manuscript.

Kind regards,

Victor Abiola Adepoju, MBCHB,Msc

Academic Editor

PLOS ONE

3. In the online submission form you indicate that your data is not available for proprietary reasons and have provided a contact point for accessing this data. Please note that your current contact point is a co-author on this manuscript. According to our Data Policy, the contact point must not be an author on the manuscript and must be an institutional contact, ideally not an individual. Please revise your data statement to a non-author institutional point of contact, such as a data access or ethics committee, and send this to us via return email. Please also include contact information for the third party organization, and please include the full citation of where the data can be found.

4. We note that you have referenced (Asparouhov T, Muthen B. Simple second order chi-square correction. Unpublished manuscript; 2010. Available from: https://www.statmodel.com/download/WLSMV_new_chi21.pdf) which has currently not yet been accepted for publication. Please remove this from your References and amend this to state in the body of your manuscript: (ie “Bewick et al. [Unpublished]”) as detailed online in our guide for authors http://journals.plos.org/plosone/s/submission-guidelines#loc-reference-style

Additional Editor Comments:

Abstract

Feedback: The abstract should mention the sample size and diversity, and briefly note the implications of the findings.

Example from Manuscript: The proposed instruments underwent content and semantic validity assessments, followed by verification of factor validity and internal consistency.

Practical Solution: Authors should revise the abstract to include a sentence like: "This study involved a diverse sample of 621 Brazilian participants from 25 different professions, indicating broad applicability of the findings. The validation of SAAS and BVI contributes to understanding self-actualization and B-values in varied Brazilian contexts, offering insights for psychological assessment and intervention."

Introduction

Feedback: Include a brief review of existing instruments measuring similar constructs to justify the need for the new instruments.

Example from Manuscript: While the introduction discusses the theoretical background, it lacks specific references to existing instruments and their limitations.

Practical Solution: Add a paragraph summarizing existing instruments, such as: "Previous instruments, such as the Personal Orientation Inventory (POI) and Characteristics of Self-Actualization Scale (CSAS), have provided foundational insights but exhibit limitations in cultural applicability and comprehensive coverage of B-values, highlighting the need for the development of the SAAS and BVI tailored to the Brazilian context."

Methods

Feedback: Provide a more detailed rationale for the choice of statistical methods.Example from Manuscript: EFA was applied to the instruments constructed in this study using the polychoric correlation matrix, with the Robust Diagonally Weighted Least Squares (RDWLS) extraction method for SAAS and Robust Unweighted Least Squares (RULS) for BVI.

Practical Solution: Expand on the choice of statistical methods by adding: The RDWLS and RULS methods were selected for their robustness in handling ordinal data and their suitability for the distribution characteristics of the survey responses, ensuring accurate factor extraction and reliability assessments in the context of psychological scale validation.

Results

Feedback: Discuss unexpected findings and their implications.

Example from Manuscript: From the analysis, one item (Item 01) was excluded due to a factor loading below 0.30 on all factors, and four items (Items 05, 31, 33, and 38) were excluded due to cross-loaded factor loadings.

Practical Solution: Provide an analysis of why these items did not perform as expected and how this affects the interpretation of the scales. For example: The exclusion of these items suggests a need to refine the conceptual alignment of the SAAS's dimensions. Future iterations could explore alternative phrasings or conceptual frameworks to ensure all aspects of self-actualization are accurately captured."

Discussion

Feedback: Offer more specific examples of how these instruments could be used in various Brazilian contexts and discuss potential cultural influences on the results.

Example from Manuscript: The SAAS fills a gap in research on self-actualization by directly linking it to the job, particularly by encompassing peak and flow experiences during the work process.

Practical Solution: Expand the discussion to include: In Brazilian cultural contexts, where job satisfaction and personal fulfillment are deeply interwoven with community and social ties, the SAAS and BVI can offer nuanced insights into how self-actualization and B-values manifest in professional settings. For instance, in education and healthcare sectors, where altruism and community service are prevalent, these instruments can help identify specific motivations and values driving job satisfaction and personal growth.

Conclusions

Feedback: Clearly state the practical applications and future research directions.Example from Manuscript: Lacks a direct statement of practical applications and detailed future research directions.

Practical Solution: Conclude with a statement like: The validation of SAAS and BVI for the Brazilian context opens avenues for their application in organizational psychology, career counseling, and personal development programs. Future research should explore longitudinal studies to assess how self-actualization and B-values evolve over time and across different life stages and professional paths.

Reviewers' comments:

Reviewer's Responses to Questions

**Comments to the Author**

1. Is the manuscript technically sound, and do the data support the conclusions?

Reviewer #1: Partly

Reviewer #2: Yes

Reviewer #3: Yes

Reviewer #4: Partly

2. Has the statistical analysis been performed appropriately and rigorously? 

Reviewer #1: Yes

Reviewer #2: Yes

Reviewer #3: Yes

Reviewer #4: No

3. Have the authors made all data underlying the findings in their manuscript fully available?

Reviewer #1: Yes

Reviewer #2: Yes

Reviewer #3: Yes

Reviewer #4: Yes

4. Is the manuscript presented in an intelligible fashion and written in standard English?

Reviewer #1: Yes

Reviewer #2: Yes

Reviewer #3: Yes

Reviewer #4: No

5. Review Comments to the Author

Reviewer #1: Thank the editors for inviting me to peer-review this manuscript. The topic is relatively meaningful and exciting. In general, the results are good and the data analyses support the instruments. I have some comments and hope that they are helpful to the authors.

1. In general, the authors divided their manuscript into numerous short parts. Several parts should be merged into longer paragraphs. A paragraph should include at least three sentences.

2. Please re-struct this manuscript following the normal structure of an article, including Introduction, Methods, Results, Discussion, and Conclusion.

3. Why did the authors choose a large number of professors (about two-thirds) for this survey? This group is only a minority of the population. Who will be the target population of these instruments?

4. Please add the information involving the response rate of participants.

5. Please explain why the authors used IPS and BVQ questionnaires in the Methods section.

6. The authors should not cite references in the Results section.

Best wishes to the authors.

Reviewer #2: General comment:

The article is well-written, however, some improvement is required to increase the clarity

Specific comment :

1. Title: The title should emphasise “the development and validation of the instrument” instead of the “measurement of the construct”.

2. Aim:

• The wording of the aim in the abstract is rather unclear. Please revise. Suggestion: Developing, validating, and cross-verifying measures for self-actualization attributes and B-values, focusing on job context and theoretical congruence with innovative behaviour and human values related to the self-actualization construct (suprapersonal subfunction).

• The wording of the aim in the introduction needs to be shortened. Please make it concise and consistent with the aim stated in the abstract.

3. Abstract:

The abstract stated that the method encompassed: “The proposed instrument underwent content and semantic validity assessments, followed by verification of factor validity and internal consistency. Additional evidence of convergent validity was also examined:” No information in the method on the constructs that instrument were cross-validated in the convergent validity and no information on the results regarding the findings related to content and validity assessment, factor analysis and internal consistency reliability

Suggestion: Rewrote the abstract to accommodate the missing information within the word limit

4. Introduction

The introduction (heading 1 ) and heading 2 is rather hard to follow, please make it more concise and simple for lay readers

Suggestion: Combine and rewrite headings 1 and 2 within the introduction section, following the IMRAD flow

5. Method

Suggestion: Following the IMRAD flow, please put Heading 3 and 4 under the method section

6. Results

Please put the findings of the content and semantic assessment in the results section instead of in the method section

Overall: Please add research implications and recommendations regarding practical and clinical application

Reviewer #3: Dear Editor,

I appreciate being given the chance to evaluate the manuscript titled “Measuring Self-Actualization and B-Values: Construction and Validation of Two Instruments in the Brazilian Context”. The authors have crafted and substantiated the validity and reliability of the Self-Actualization Attributes Scale (SAAS), a tool assessing self-actualization; they have constructed and established the validity and reliability of the B-Values Inventory (BVI), which pertains to B-values; and they analyzed the convergence between the factors of the SAAS and the BVI. In the manuscript, the authors not only provided a comprehensive review based on the underlying constructs and variable operationalization, but clearly described the content and semantic validity procedures, and conducted careful psychometric analyses, including exploratory factor analysis, quality of fit, predictive effectiveness, reliability, factor determination, parallel analysis, internal consistency, Cronbach's alpha, composite reliability, McDonald’s Omega, factor stability, and others. In addition to that, the authors attached crucial material that states the ethical standards as well as the context where the project was submitted. The results are two interesting scales that certainly fill a gap in research on self-actualization and being valued in the light of Maslow’s motivation theory. As an experienced reviewer, I may register that it is not often that we receive a manuscript so attempted in the details. Ideally, if possible, I would have expected some additional criteria validity analysis, assessing the extent to which a person's performance on these questionnaires conforms to external criteria or standards that are relevant to the construct the test is intended to measure, such as burnout, stress, or some general mental health. If the authors have something like that, I suggest including it. Nevertheless, this would be a minor suggestion.

Reviewer #4: ID: PONE-D-24-00059

Title: Measuring Self-Actualization and B-Values: Construction and Validation of Two Instruments in the Brazilian Context

Thank you for providing a chance to review this manuscript.

Abstract

1) Purpose and results are not adequately presented, which results in the reader not being able to extract the focus of the study from it, and the authors are asked to add clarity.

2) It is recommended that authors use subheadings to make the abstract section clearer, as in the case of “Background”, “Objective”, “Methods”, “Results”, “Conclusion”.

Overall: Abstracts require a brief summary of the article's background, aims, methods, results and conclusions. The current abstract does not provide a good overview. Please refer to abstracts of high quality articles and rewrite the abstract.

Introduction

Line34-37 Page 1: Please describe each requirement specifically.

Line52-56 Page 1: Please elaborate further. The current description does not give me a clear picture of this part of your research.

Methods

Line229-233 Page 6:

1) Whether there were inclusion or exclusion criteria for the study population, and if so, please state this in the article.

2) How was the sample size determined, give the calculation or supporting references.

3) Is 621 the final study sample?

4) Could the authors please clarify, as well as whether there is missing data and how to deal with missing data.

5) It is recommended that a table of demographic characteristics be created.

6) “From 25 different 229 professions”, please indicate what each of the 25 occupations is?

Line 235-238 Page 6: Please describe the process in detail, e.g. what is the source of the participants? Are participants paid accordingly, etc.?

Line 286-287 Page 7: “ v12.01.02 286 ”, “v” is an abbreviation, used here for the first time, which should be interpreted (version), Please revise it.

Results

Line 319 Page 8: Authors are asked to double-check the parts of the text that require italics, such as χ2.

Line 355 Page 10:

1) The decimal points are not harmonized.

2) The author needs to check that all abbreviations are explained e.g. CFI, TLI.

3) (χ2) needs to be italicized.

Line 367 Page 10: “(χ2(91) 367= 3052.8; p = 0.00)”, χ2 needs to be italicized.

Line 379 Page 11: “(H =0.87)”, Spaces are required before and after the equal sign.

Line 387 Page 11:

1) The decimal points are not harmonized.

2) The CFI TLI is not explained.

3) (χ2) needs to be italicized.

Conclusion

Overall: For an article, the conclusion is a necessary demonstration of important findings and the value of the research, which the authors are invited to add.

Other recommendations:

Overall: Give the source of funding and the role of the funders for the present study.

The author does not seem to be very good at essay writing and many of the basic requirements are not met. There are a number of problems with this essay: 1) punctuation conforms to incorrect use, 2) much of it is not detailed enough, and 3) the current analysis is too simplistic to explain the purpose of the study.

Thank you and my best,

Your reviewer

6. PLOS authors have the option to publish the peer review history of their article (what does this mean?). If published, this will include your full peer review and any attached files.

Reviewer #1: No

Reviewer #2: **Yes: **Novita Intan Arovah

Reviewer #3: **Yes: **Bruno Kluwe-Schiavon

Reviewer #4: No

---

## [Author Response · Author response to Decision Letter 0]

29 Feb 2024

Dear Editor,

We would like to inform you that we reviewed all the points indicated in the general editorial review. And all the points indicated by the reviewers have been taken into account. We have reviewed each point of correction, adequacy, or insertion. We would like to point out that some points were only partially resolved; however, these points were duly and thoroughly justified, as shown in the table below.

We remain at your disposal for any further revisions.

Best regards,

Prof. Gustavo Henrique Silva de Souza

Prof. Jorge Artur Peçanha de Miranda Coelho

Prof. Nilton Cesar Lima

Prof. Germano Gabriel Lima Esteves

Prof. Fernanda Cristina Barbosa Pereira Queiroz

Prof. Yuri Bento Marques

---

## [Editor Report · Decision Letter 1]

2 Apr 2024

Self-Actualization and B-Values: Development and Validation of Two Instruments in the Brazilian Context

PONE-D-24-00059R1

Dear Gustavo Henrique Silva de Souza and Colleagues,

We’re pleased to inform you that your manuscript has been judged scientifically suitable for publication and will be formally accepted for publication once it meets all outstanding technical requirements. After a detailed review of the revised manuscript "Self-Actualization and B-Values: Development and Validation of Two Instruments in the Brazilian Context" by Gustavo Henrique Silva de Souza and colleagues, and considering the authors' comprehensive response to feedback, the study's innovative contribution to the field, and the rigorous methodological enhancements made, the final decision is to ACCEPT the manuscript for publication in PLOS ONE.

Kind regards,

Victor Abiola Adepoju, MBCHB,Msc

Academic Editor

PLOS ONE

Additional Editor Comments (optional):

Dear Gustavo Henrique Silva de Souza and Colleagues,

Congratulations on the successful revision of your manuscript titled "Self-Actualization and B-Values: Development and Validation of Two Instruments in the Brazilian Context." Your efforts to address the reviewers' comments have significantly enhanced the clarity, depth, and contribution of your work. The revisions made, particularly in strengthening the methodological rigor and expanding on the theoretical implications of your findings, have positioned your article as a valuable contribution to the fields of psychology and self-actualization research.

Your work addresses an important gap in the literature by offering robust instruments for assessing self-actualization and B-values within the Brazilian context. This contribution not only advances our understanding of self-actualization but also provides practical tools for further research and application in this area.

We are pleased to inform you that your manuscript has been recommended for acceptance for publication in PLOS ONE. We believe your research will be of great interest to our readership and look forward to its publication.

Best regards,
---

## [Editor Report · Acceptance letter]

13 May 2024

PONE-D-24-00059R1 

PLOS ONE

Dear Dr. Souza, 

I'm pleased to inform you that your manuscript has been deemed suitable for publication in PLOS ONE. Congratulations! Your manuscript is now being handed over to our production team.

Kind regards, 

on behalf of

Dr Victor Abiola Adepoju 

Academic Editor

PLOS ONE